# Physiological and Biochemical Characteristics of Rainbow Trout with Severe, Moderate and Asymptomatic Course of *Vibrio anguillarum* Infection

**DOI:** 10.3390/ani12192642

**Published:** 2022-10-01

**Authors:** Stanislav Rimaso Kurpe, Irina Viktorovna Sukhovskaya, Ekaterina Vitalyevna Borvinskaya, Alexey Anatolievich Morozov, Aleksey Nikolaevich Parshukov, Irina Evgenyevna Malysheva, Alina Valeryevna Vasileva, Natalia Alexandrovna Chechkova, Tamara Yurevna Kuchko

**Affiliations:** 1Institute of Biology, Ecology and Agricultural Technologies of the Petrozavodsk State University (PetrSU), 185640 Petrozavodsk, Russia; 2Institute of Protein Research of the Russian Academy of Sciences, 142290 Pushchino, Russia; 3Institute of Biology of the Karelian Research Centre of the Russian Academy of Sciences (IB KarRC RAS), 11 Pushkinskaya Street, 185910 Petrozavodsk, Russia; 4Institute of Biology, Irkutsk State University, 664003 Irkutsk, Russia; 5Limnological Institute of the Siberian Branch of the Russian Academy of Sciences (LIN SB RAS), 3 Ulan-Batorskaya Street, 664033 Irkutsk, Russia

**Keywords:** aquaculture, infection, plasma proteomics, rainbow trout, stress response, *Vibrio anguillarum*, vibriosis

## Abstract

**Simple Summary:**

During the past decades, bacterial infections have been a serious problem in aquaculture that causes very large economic losses. Currently, antibiotics are the most common method of disease prevention and control. A combination of water quality monitoring, early detection of fish infections, and other preventive biosecurity measures in fish farms can help prevent the spread of infection. We investigated the natural bacterial infection in fish farms and characterized the parameters of the health status of rainbow trout Oncorhynchus mykiss (Walbaum, 1792) during disease. Depending on the course of the disease (severity of the pathology, leukocyte profile, and expression of immune-related genes), three subpopulations of fish with severe damage, a moderate course of the infectious process, and asymptomatic fish were characterized. An unexpected result was a small metabolic difference between fish with moderate symptoms and fish with weak signs of pathology. Thus, we have described the characteristics of a trout subpopulation with a mild course of infection which has potential for recovery after infection.

**Abstract:**

This article describes the clinical manifestation of natural *Vibrio anguillarum* infection in rainbow trout (Oncorhynchus mykiss) during an outbreak on a fish farm. (i) Using an integrated approach, we characterized the pathogenesis of vibriosis from the morphological, hematological, and biochemical points of view. The molecular mechanisms associated with the host immune response were investigated using mass spectrometric analysis of trout plasma proteins. (ii) According to the severity of infection (the extent of tissue damage, the level of expression of pro-inflammatory genes, and changes in the leukocyte profile) three fish populations were identified among infected trout: fish with severe lesions (SL), fish with the moderate infectious process (IP) and asymptomatic fish (AS). (iii) Lymphopenia, granulocytosis, and splenomegaly were strong trends during the progression of infection and informative indicators of severe manifestation of disease, associated with hemorrhagic shock, metabolic acidosis, and massive tissue damage. (iv) As expected, pro-inflammatory interleukins, complement components, acute phase proteins, and antimicrobial peptides were implicated in the acute pathogenesis. Systemic coagulopathy was accompanied by increased antithrombotic reactions. (v) Reconstruction of metabolic pathways also revealed a high energy requirement for the immune response in severely affected fish. (vi) An unexpected result was a small difference between fish with moderate symptoms and fish with no or minor external signs of pathology (putatively resistant to infection). Increased production of antiproteases and enhanced blood coagulation cascade were observed in healthier fish, which may underlie the mechanisms of a controlled, non-self-damaging immune response to infection. (vii) Depending on the progression of the disease and the presence of the pathogen, a stepwise or linear change in the abundance of some plasma proteins was revealed. These proteins could be proposed as molecular markers for diagnosing the health and immune status of trout when cultured in fish farms.

## 1. Introduction

Rainbow trout is one of the most common cultivated fish species in the world, grown in cold, fresh, or, less often, brackish waters [1]. The capacity of the world’s inland cold water ecosystems for aquaculture is limited and rapidly depleting because fish farming leads to rapid pollution of water bodies with organic substances [2]; therefore, sustainable development of aquaculture requires more intensive farming in estuaries and on the seashore. However, when reared in brackish water, trout are at high risk of being infected with halophilic bacteria *Vibrio* sp., an extremely infectious, rapidly developing, and very deadly pathogen that causes large economic losses in aquaculture [3]. Therefore, the development of scientifically based protection measures against this infectious agent is very important for trout farming.

To date, many studies have shown high variability of individual resistance to *Vibrio anguillarum* infection in fish [4], but the mechanisms of this selective susceptibility are still being elucidated. There is a lot of information in the literature about the cellular and tissue response of fish to vibriosis, while information about the biochemical and molecular mechanisms underlying the individual dynamics of the disease is still very limited [4,5] The recent detailed molecular studies have shown the importance of expression of genes encoding immunoglobulins, antimicrobial peptides and regulating various types of T cells for the survival of fish [6] and revealed the phenotype-specific features of the complement and coagulation cascades and TNF-associated regulation in resistant fish [4,7].

The circulatory system is one of the key body compartments for studying the mechanisms of *Vibrio* pathogenesis. It is primarily affected by *V. anguillarum*, since this infection typically manifests as anemia and a septic hemorrhage, probably provoked by bacterial endotoxins [8,9]. Vibriosis is also characterized by the destruction of the vascular endothelium and the most extensive histopathologies of blood vessels and hematopoietic tissues [10]. At the same time, blood provides a cellular and humoral immune response that determines the course of the disease and mediates the focus of infection with the lymphoid tissues. However, with a few exceptions, there is a lack of molecular studies of *V. anguillarum* infection focused on the blood system [11].

In this work, we aimed to characterize the clinical signs of infectious processes caused by *V. anguillarum* in cultured trout, including alterations in the composition of plasma proteins and populations of blood cells. At the molecular level, the pathogenesis of natural *V. anguillarum* infection has not previously been studied in the field, where the variation in bacterial loads gives particular individuals a chance to overcome the disease at the early stages, in contrast to experiments where an effective pathogen titer is deliberately used [4,12,13]. Therefore, the article discusses a possible relationship between the production of certain blood proteins in individual fish, their hematological profile, physiological state, and the severity of the infection process. We believe that the detailed information obtained on the progression of disease caused by *V. anguillarum* in cultured trout will be useful for the search for molecular targets for the treatment of vibriosis and genetic selection of disease-resistant breeds. In this work, for the first time, we have obtained detailed data on the individual response of rainbow trout to natural infection using an integrated approach that includes a combination of methods from ichthyology, physiology, and molecular biology.

## 2. Materials and Methods

### 2.1. Animal and Water Collection

The fish was caught in August 2020 on a trout farm located in the bay of the White Sea, Northwest Russia. On this cage farm, fish are cultivated in brackish water in summer (12–14‰), and in the estuary of the nearest river in winter. The fish were caught by a net randomly from four neighboring cages. The animals in these cages were from the same full-sibling family batch, the same age, average weight, and were kept with the same stocking density and feeding regime (Appendix A).

Different groups of fish with obvious symptoms of infection (red fins, visible surface ulcers, and hemorrhagic lesions, pale gills, exophthalmia) and those not showing any outward signs of illness were caught from the cages on the trout farm where there was an outbreak of vibriosis (Figure 1). As a result, a total of 19 trout (12 individuals with visual signs of infection and 7 relatively healthy fish) with an average weight of 1827 ± 471 g (mean ± SE) were sampled (Appendix A). All trout were females, so gender was not taken into consideration in further analysis. Each fish was euthanized in a 0.5 mL/L clove oil emulsion for about 5 min until it stopped responding to the pinching of the fin with tweezers. If there were skin lesions, the mucus was scraped from the affected area with a sterile scalpel and fixed in liquid nitrogen for 16S rRNA metabarcoding analysis. Then the weight and length of the fish were measured, and the external signs of pathologies of the skin and gills were recorded in the table of ichthyopathological examination (Appendix A). After that, the fish was dissected aseptically, and blood was taken from the heart using a syringe treated with 3.6% sodium citrate as an anticoagulant. Further, the anomalies of the internal organs were visually assessed, and also were added to the ichthyopathological table. Photographs of the external integument and internal organs of each fish were also made (available in https://doi.org/10.6084/m9.figshare.17169380, accessed on 13 December 2021). The spleen and liver of the fish were weighed, then fixed in liquid nitrogen for further analysis by PCR and 16S rRNA gene sequencing. The formula was used to calculate the relative weight of the spleen or liver (spleen somatic or hepatosomatic index, respectively).
$$I=\frac{O}{F}$$
where

I—relative weight of organO—organ mass, gF—fish mass, g

**Figure 1 animals-12-02642-f001:**
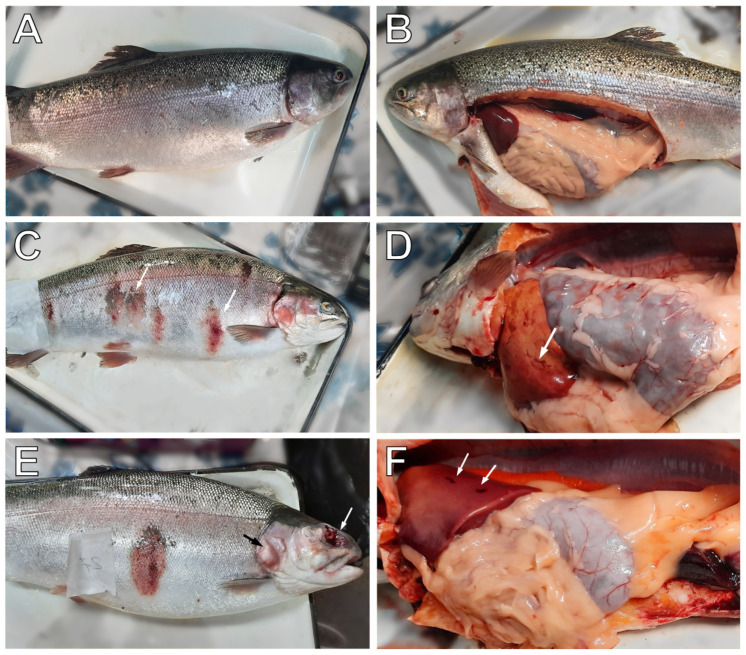
Representative anomalies of rainbow trout collected from a fish farm during a vibriosis outbreak with varying degrees of infection. (**A**)—asymptomatic fish with insignificant signs of disease on the skin and fins (AS group). (**B**)—dissection of asymptomatic fish with no hemorrhages in the body cavity (AS group). (**C**)—fish with “red marks” on the skin (white arrows) indicating infection (IP group). (**D**)—local hemorrhages (white arrow) in the liver of an infected trout (IP group). (**E**)—blisters (black arrow) and complete eye loss (white arrow) in fish with severe skin lesions and ulcers (SL group). (**F**)—hemorrhagic foci (white arrow) in the liver of severely affected fish (SL group).

The 10 L (total) water samples near the water surface at a distance of one meter from the cage were collected by using two sterile glass bottles (5 L × 2) and immediately filtered through a sterile 0.22 μm filter cartridge. The filter was transferred in a cryovial, frozen, and stored in liquid nitrogen until DNA extraction.

### 2.2. Fish Pathology Characterization (Diagnostic Criteria)

To rank fish (*n* = 19) welfare according to the severity of the infectious process, a semi-quantitative scoring system was used. For this, each observed pathological change in skin, gills, liver, spleen, muscles, kidney, gastrointestinal tract, etc., of an individual fish was assigned points of ichthyopathology (1 point corresponded to the normal state, 2 points to weak pathology, and 3 points to severe pathology). The obtained points were summed up, and an integral score of ichthyopathology (IPS) was obtained for each fish (Appendix A).

### 2.3. Hematological Parameters

The hematologic profile was partially characterized using blood smears, made immediately after fish blood collection (*n* = 19) according to a standard protocol [14]. Romanowsky-type staining was applied to differentiate cells in microscopic examinations. Air-dried and fixed in ethanol (5 min) smears were immersed in May–Grünwald solution (MiniMed, Russia) for 3 min, rinsed with water, dried, and dipped for 45 min in DiaChim-Hemistain-R solution (“NPF ABRIS+”, Russia) in PBS buffer, pH 6.8–7.2. Then they were rinsed in distilled water, dried, and examined using the «Motic» microscope with «Moticam» digital camera with the device «MI Devices». Blood cell types were recognized according to hematological atlases for fish [15,16,17]. The differential counts of blood cells were performed in homogeneous areas by counting 500 cells on each smear. The relative number of erythrocytes and thrombocytes was expressed as a percentage of all counted cells (Appendix A). The relative number of leukocytes was obtained by subtracting the percentage of thrombocytes from the total of white blood cells. The relative numbers of lymphocytes, monocytes, and granulocytes were expressed as % of total leukocytes (Appendix A).

### 2.4. Expression of Immune-Related Genes

A spleen sample (*n* = 19) weighing about 50 mg was homogenized in 1 mL of PureZOL™ RNA isolation reagent (Bio-Rad, Hercules, CA, USA), followed by the addition of 200 μL of chloroform. Then, tissue debris was pelleted by centrifugation at 12,000 rpm for 5 min at +4 °C on a Allegra 64R centrifuge, (Beckman Coulter, Brea, CA, USA). The supernatant was transferred to a new tube and mixed with an equal volume of isopropyl alcohol.

Then the samples were centrifuged again at 4 °C under the same conditions, and the precipitate was washed twice with 80% ethanol and dried for 5 min, followed by dissolving in 100 μL of deionized sterile water. Residual genomic DNA was cleaved by DNase enzyme (1 U) at 37 °C for 30 min (Sibenzyme, Russia). RNA quantity and integrity were estimated with agarose gel electrophoresis.

Complementary DNA was synthesized using the “MMLV RT” kit (Evrogen, Russia) according to the recommendation of the manufacturer. qPCR was performed with iCycler iQ5 (Bio-Rad, USA) with Screen-Mix SYBRGreen reagents (Evrogen, Russia) with the following cycling conditions: denaturation temperature was set to 95 °C for 3 min (initial denaturation) or for 30 s (in each cycle); annealing temperature was set to 58 °C for 30 s, and elongation temperature was set to 72 °C for 30 s. The absence of non-specific product bands was confirmed with melting curve analysis and agarose electrophoresis. Amplification efficiency was tested and lay within the 98–100% range for all primer pairs. The level of gene transcripts was calculated from ΔΔCt [18]. RT-PCR for each sample was performed at least three times. Primer sequences for genes of immune response were taken from literature or generated in the Beacon Designer 5.0 (Appendix A) [19,20,21]. The expression of the immune genes was normalized to the level of ß-actin as reference (Appendix A).

### 2.5. Grouping Fish according to the Severity of the Disease

An integrated approach was applied to classify collected fish based on all measured physiological and biochemical traits (morphometry, hematological profile, and immune-related gene expression). To cluster and visualize the differences between experimental groups, PCA analysis and hierarchical clustering on principal components (HCPC) were performed in the ‘factoextra’ package for R. As a result, three groups of fish were assigned for further analysis: asymptomatic (AS), with infectious process (IP), and with severe lesions (SL) (Figure 2A).

Cluster analysis revealed a subgroup of fish clearly distinguishable among the fish collected from the fish farm: these fish had a larger relative size of the spleen, a lower proportion of lymphocytes, a higher proportion of granulocytes in the blood, and the highest level of expression of the pro-inflammatory cytokines il1b and il8 (Appendix A). These fish with the pathophysiology of systemic inflammatory response syndrome (SIRS) and secondary organ damage identified by cluster analysis were designated as the “severe lesions” group (SL), and their plasma samples (*n* = 5) were taken for proteomic analysis. The rest of the fish collected in the fish farm constituted one rather homogeneous group with a similar level of immune response parameters, which included both fish with obvious signs of infection (hereinafter “infectious process (IP) group) and individuals looking relatively healthy (hereinafter “asymptomatic” (AS) group). Some of the fish from these groups were selected for plasma proteome assay (*n* = 4, both), while the rest were not used for further analysis (Figure 2A).

The relationship between the studied physiological, morphological, and biochemical parameters was assessed using the Spearman correlation in the ‘corrplot’ package for R (Figure 2B). The expression level of immune response genes, the relative numbers of white blood cell populations, and morphometric parameters of trout were compared between fish groups using Wilcoxon’s two-sample test with Benjamini-Hochberg correction for multiple testing using ‘coin’ and base stats packages for R; *p* < 0.05 were considered statistically significant (Figure 2C–F).

### 2.6. Identification of the Pathogen Using 16S rRNA Gene Amplicon Sequencing

Total DNA was isolated from fish spleen and skin lesions. The obtained OTUs are available at https://doi.org/10.17632/6m4jz3ywhc.2 (accessed on 13 December 2021). Briefly, V3–V4 variable regions of the 16S rRNA gene were amplified and the resulting pool of libraries was sequenced on an Illumina MiSeq (2 × 300 bp paired-end reads). The resulting sequences were used to generate 97% identity OTUs using mothur 1.44.11 software and then aligned to NCBInr using the megaBLAST web server. The composition of communities was plotted using the ‘*phyloseq*’ R package. To explore the overall relationship of communities, NMDS plots were built using the Bray–Curtis distance measure.

### 2.7. Plasma Proteome Analysis

Plasma proteome analysis was performed using the equipment of the “Human Proteome” Core Facility of the Institute of Biomedical Chemistry (Moscow, Russia). Freshly collected whole blood samples (*n* = 13) were transferred into microtubes and centrifuged at 5000 g, 4 °C, for 10 min. Plasma was collected, frozen in liquid nitrogen, and stored at −80 °C. On the day of analysis, plasma samples were thawed, and the protein concentration was measured using a Pierce™ BCA Protein Assay Kit (Pierce, Rockford, IL, USA). For this, 1 mL of a reagent containing 1% sodium salt of bicinchoninic acid, 2% Na_2_CO_3_, 0.16% sodium tartrate, 0.4% NaOH and 0.95% NaHCO3 (pH = 11.25) and 20 μL of a 4% CuSO4 was added to 30 μL of the sample. Then the solutions were mixed and incubated at 56°C for 20 min on a Termomixer comfort shaker (Eppendorf, Germany). The concentration of soluble protein was determined at a wavelength of 562 nm on a NanoDrop ND-1000 spectrophotometer (Thermo Fisher Scientific, Waltham, MA, USA) using a calibration curve with standard BSA solutions (3 replicates).

A total protein amount of 100 mg of each sample was used for tryptic digestion according to the common FASP protocol [22]. Protein disulfide bridges were reduced in 10 mmol/L 1,4-dithiothreitol in 100 mmol/L Tris-HCl (pH 8.5) for 20 min at 56 °C. Alkylation of thiols was performed in 55 mmol/L iodoacetamide (Sigma-Aldrich), 8 mol/L urea in 100 mM Tris-HCl (pH 8.5) at room temperature for 30 min in the dark. Tryptic digestion was carried out overnight at 37 °C with trypsin (Sequencing Grade Modified, Promega; protein/trypsin ratio 1:50) in 8 mM tetraethylammonium bicarbonate buffer (pH 8.5). To stop the hydrolysis, formic acid was added to a final concentration of 5%, followed by centrifugation at 11,000 g for 15 min. The obtained supernatant was dried in a vacuum concentrator Concentrator 5301 (Eppendorf) and dissolved in 20 mL of 0.1% formic acid to a final concentration of 1 µg/µL.

Proteome analysis was performed using an Ultimate 3000 Nano LC System (Thermo Scientific, Rockwell, IL, USA) connected to a Q Exactive HF-X Hybrid Quadrupole-OrbitrapTM Mass spectrometer (Thermo Scientific, Waltham, MA, USA) using the method described below. A volume of 1 µL peptides was loaded onto the Acclaim µ-Precolumn (0.5 mm × 3 mm, 5 µm particle size, Thermo Scientific) at a flow rate of 10 µL/min for 4 min in an isocratic mode of Mobile Phase C (2% acetonitrile (Sigma-Aldrich), 0.1% formic acid). Then the peptides were separated with high-performance liquid chromatography (HPLC, Ultimate 3000 Nano LC System, Thermo Scientific, Rockwell, IL, USA) in a 15 cm long C18 column (Acclaim® PepMap™ RSLC inner diameter of 75 µm, Thermo Fisher Scientific). The peptides were eluted with a gradient of buffer A (0.1% formic acid) and buffer B (80% acetonitrile, 0.1% formic acid) at a flow rate of 0.3 μL/min. The total run time was 90 min, which included an initial 10 min of column equilibration to 2% buffer B, then a gradient from 2 to 35% of buffer B over 68 min, then 2 min to reach 99% of buffer B, flushing 2 min with 99% of buffer B, and a linear decrease in the concentration of buffer B to the original 2% in 3 min and 5 min re-equilibration.

Mass spectrometry analysis of the samples was performed at least in triplicate with a Q Exactive HF-X mass spectrometer in positive ionization mode using an nESI ion source. The temperature of the capillary was 240 °C, and the voltage at the emitter was 2.1 kV. Mass spectra were acquired at a resolution of 120,000 (MS) in a range of 300−1500 m/z. Tandem mass spectra of fragments were acquired at a resolution of 15,000 (MS/MS) in the range from 100 m/z to m/z value determined by a charge state of the precursor, but no more than 2000 m/z. Isolation of precursor ions was performed in ±1 Da window. Up to 40 ions was set as the maximum number of ions allowed for isolation in MS2 mode. For tandem scanning, only ions with a charge state from z = 2+ to z = 6+ were taken. The maximum integration time was 50 ms and 110 ms for precursor and fragment ions, respectively. AGC targets for precursor and fragment ions were set to 1 × 10^6^ and 2 × 10^5^, respectively. An isolation intensity threshold of 50,000 counts was determined for precursor selection and the top 20 precursors were chosen for fragmentation with high-energy collisional dissociation (HCD) at 29 NCE. All measured precursors were dynamically excluded from triggering a subsequent MS/MS for 90 s. All mass spectrometry data are deposited in ProteomeXchange with the identifier PXD031356.

The mass spectra raw files were loaded into the MaxQuant v.1.6.4.3 program. The searches were performed using the Andromeda algorithm (built into MaxQuant) using the *Oncorhynchus mykiss* database provided by UniProt (April, 2021). The following search parameters were used: enzyme specificity was set to trypsin, and two missed cleavages were allowed. Carbamidomethylation of cysteines was set as a fixed modification, and methionine oxidation and N-terminal proteins acetylation were set as variable modifications for the peptide search. The mass tolerance for precursor ions was 4.5 ppm; the mass tolerance for fragment ions was 20 ppm. Peptide Spectrum Matches (PSMs), peptides, and proteins were validated at a 1% false discovery rate (FDR), estimated using the decoy hit distribution. Proteins were considered to be significantly identified if at least two peptides were found for them.

Protein quantification was based on the label-free quantification (LFQ) method. The obtained LFQ intensities are available in Appendix A. The data were filtered to exclude proteins identified by modified (only identified by side) and reverse peptides, potential contaminants, and proteins with a Score < 25 and with no unique peptides (Appendix A). Before analysis, basic filtering was carried out, when samples containing identified protein in only one technical repeat were considered protein-free (single value replaced by “NA”). Then data were log2-transformed and normalized by median centering method using the ‘*limma*’ package for R. Then for each sample, the median of the technical replicas was used. The imputation of missing values procedure was performed by the MinProb method using ‘*imputeLCMD*’ packages for R. The obtained data matrix can be found in Appendix A.

The comparison of plasma protein expression in the studied groups of fish was performed using the linear modeling in the ‘limma’ package for R (*p* < 0.05). Differentially expressed proteins were clustered based on Spearman’s correlations between their LFQ values in all samples. Obtained clusters of co-expressed proteins performed in Appendix A and plotted on a heatmap through the ‘heatmap.2′ function in the ‘gplots’ package v.2.17.0.

The functional annotation of identified rainbow trout proteins was analyzed using the BlastKOALA and PANNZER2 services. The reconstruction of metabolic pathways was conducted by tools of the Kyoto Encyclopedia of Genes and Genomes (KEGG). KEGG pathway maps were generated based on functional orthologs of plasma differentially expressed proteins using the ‘*pathview*’ package for R. Gene Ontology term enrichment was examined using the “topGO” package for R.

## 3. Results

### 3.1. Clinical Examination of Collected Fish

In diseased fish from the trout farm, the most common symptoms of infection were hyperemic inflammatory skin lesions (“red spots”), with massive scale loss (Figure 1, Appendix A, photographs supporting Appendix A are available at https://doi.org/10.6084/m9.figshare.17169380, accessed on 13 December 2021). Severely affected individuals with swelling and abscesses in the muscles and on the head were observed. In the terminal stage, these skin abscesses burst, leaving deep wounds. A less common symptom of infection was a hyperemic and protruding anus, indicative of intestinal inflammation. Swollen orbits (exophthalmia) and small foci of hemorrhages in the liver, reported as a clinical symptom of vibriosis, were also found in some individuals [23,24].

Asymptomatic trout with normal skin color had no swellings or spots on the surface. Scales do not stick out, although many individual scales were accidentally lost, and minor petechial hemorrhages on the skin, fins, and in the eyes or slight swelling of the anus could be occasionally observed (Appendix A). All farmed fish had slight fin necrosis, which is typical at high fish stocking density.

Correlation analysis showed the relationship between the severity of pathology and the applied biomarkers of the immune response (immune-related genes, relative numbers of white blood cells, and types of leukocytes in the blood; Figure 2B). The level of tissue damage for each fish, expressed as the total score of external signs of ichthyopathologies (IPS), was positively correlated with spleen size and the proportion of granulocytes (r = 0.6 and 0.5, respectively). The size of the spleen (lymphoid organ) reflected the activation of cellular immunity in infected trout: the larger the spleen, the greater the relative number of granulocytes and the fewer lymphocytes (r = 0.7 and −0.9, respectively). In general, the proportion of lymphocytes was negatively correlated with the proportion of granulocytes in the blood (r = −0.8). The levels of expression of proinflammatory genes interleukin *il1b* and *il8* in the spleen were positively correlated with IPS (r = 0.5 and 0.4, respectively) and were strictly coordinated with each other (r = 0.8). In turn, the expression of anti-inflammatory cytokine transforming growth factor β1 (*tgfb*) did not correlate with the severity of external signs of pathology or other measured indicators of the fish welfare status.

### 3.2. Molecular Identification of Pathogen

To determine the causative agent of the disease, skin lesions (if there were any) and the spleen of fish were analyzed by 16S rRNA gene sequencing. Overall, 85 non-singleton OTUs were detected in the samples (https://doi.org/10.17632/6m4jz3ywhc.2 (accessed on 13 December 2021)). Bacterial communities in the wounds and skin lesions were dominated by *Vibrio* spp. (Figure 3B); *V. anguillarum* was identified as one of three major OTUs representing this genus. Between these three OTUs (which most likely represent a single strain), *Vibrio* sp. reached 98% of total bacterial abundance in some wound samples. Other bacteria in the libraries obtained from wounds were probably a mixture of opportunists and random elements of the background environmental community; none of these bacteria were present in all infected fish, and some of them were shared with the background water community.

Spleen is an important immune organ of fish, which accumulates bacteria that enter the body [25]. *Vibrio*-rich libraries were more often found in the spleen of trouts from the SL group, indicating serious bloodstream infection, while these OTUs were also sporadically found in the spleen of both asymptomatic and moderately infected fish (Figure 3A). Libraries without significant *Vibrio* abundance were also produced from fish in all three groups. It is important to note that most libraries had very few reads (Appendix A), while those with the highest overall abundance (specimen 23, 28, 36, and 37) were all dominated by Vibrio OTUs.

### 3.3. Proteomic Analysis

Comparative analysis of the blood plasma proteins of trout was performed with the method of highly efficient bottom-up proteomics. As a result, 532 proteins were reliably identified in 13 studied samples; 231 of them were filtered for downstream analysis of differential expressions. Using the Limma linear model, 46 proteins were identified with the LFQ approach, which were differentially expressed between the three experimental groups of fish.

The largest differences in blood plasma composition were found between the severely diseased individuals (SL) and two other fish groups: 32 proteins were differentially expressed (DEPs) in comparison with an asymptomatic fish (AS) and 38 proteins in comparison to fish with the moderate infectious process (IP) (Figure 4A,B); about half of these DEPs were shared by the AS and IP groups. Between the AS and SL groups, eight unique proteins were differentially produced, while a comparison between the IP vs SL groups revealed 14 unique DEPs. At the same time, differences between fish from the AS and IP group were negligible (1 single plasma DEP cadherin; A0A060WUV2). 

Two groups of proteins with coordinated expression were revealed with correlation analysis in fish plasma proteome (Figure 4C). The largest group (cluster 1) included proteins that were abundant in the plasma of heavily infected fish (SL fish) and were positively correlated with the expression of *il8* and *il1b* and granulocyte proportion in blood cells (Appendix A). This group comprised the antimicrobial peptides (lysozyme g and C II (C1BGU6 and P11941, respectively), ceruloplasmin (A0A060Y0R7), and cathelicidin (Q08G23)) and markers of the acute phase of inflammation (precerebellin-like protein (Q9PT14), haptoglobin (A0A060XGG8 and A0A060YZR6), intelectin (P0DMV4), apolipoprotein M (A0A060XKK4)) and apolipoprotein M-like protein (A0A060YST0), leukocyte cell-derived chemotaxin 2 (C1BG12) and lymphocyte cytosolic protein plastin (A0A060XLW3)). Components of the complement system (complement component C7-1 (A0A060WHT8) and complement factor B-like (A0A060X8X3)), as well as coagulation cascade proteins (fibrinogen (A0A060XUK2 and A0A060YWY3), angiotensin (A0A060WGU6) and pigment epithelium-derived factor (A0A060X9D3)), were also abundant in this group of trout proteins. Along with pro-inflammatory molecules, cluster 1 comprised plasma proteins that reduce inflammation and complement activation, such as the actin depolymerization factor gelsolin (A0A060WRB4), CD59 glycoprotein-like protein (Q9DFD5), and catechol-O-methyltransferase domain-containing protein (A0A060YXK5 and A0A060Z9B8). Tissue damage markers creatine kinase (A0A060WKG4 and S0F2P2), hemorrhage-associated enzyme nucleoside diphosphate kinase (A0A060WE68), and battery of intracellular glycolytic enzymes (phosphoglycerate mutase (A0A060Z9H8), triosephosphate isomerase (A0A060VUA9), fructose-bisphosphate aldolase (A0A060WUI1, A0A060VVE5), glyceraldehyde-3-phosphate dehydrogenase (Q90ZF1), phosphoglycerate mutase (A0A060Z9H8), phosphoglycerate kinase (A0A060WC08), enolase (A0A060YWQ3) and L-lactate dehydrogenase (A0A060WGR2)) were also widely represented in this cluster. Therefore, proteins from cluster 1, increased in seriously affected fish, are presumably direct and mediated products of the immune reaction cascade and related cytokine-mediated damage to fish tissues.

The second cluster of DEPs consisted of proteins decreased in their abundance in SL fish, which were negatively correlated with the relative number of granulocytes and *il8* expression. This result may indicate their role in anti-inflammatory reactions. These included a range of apolipoproteins: apolipoproteins M (isoform A0A060YU50), apolipoprotein A-I-like (A0A060YKT0), and apolipoproteins B-100-like (A0A060WCH3 and A0A060W754). In plasma of fish from the SL group, coagulation factor kininogen (A0A060WI78) and prothrombotic antiproteases (three isoforms of alpha-2-macroglobulin (A0A060YZ84, A0A060WUN3, and A0A060WW01) and inter-alpha-trypsin inhibitor heavy chain H3-like (A0A060XRC5)) were also significantly decreased compared to the healthier fish. Immunoglobulin light chain (kappa) precursor (A0A060WXT1) and mediator of inflammation pentraxin (P79899) with immunoglobulin-like action were also decreased in severely affected fish, as well as plasma hemoglobin subunit alpha (A0A060YJC7), despite the supposed hemolytic effect of bacterial toxins [25].

Based on Gene Ontology classification, most up-/down-regulated DEPs were related to such GO terms as peptidase activity, carbohydrate binding, small molecule metabolic process, generation of precursor metabolites and energy, protein processing, response to wounding, blood coagulation, and negative regulation of blood coagulation and regulation of response to stress (Appendix A). This reflects immune reactions, extensive hemolytic processes, and cellular and endothelial damage associated with *Vibrio infection*.

The presence of serine-type endopeptidases, as well as proteins with negative regulation of endopeptidase activity, is also noticeable among trout DEPs, indicating an important role of proteolytic enzymes in inflammation progression. A remarkable abundance of apolipoproteins among both up- and down-regulated DEPs causes a noticeable increase in the frequency of lipid transporter GO term; however, in infected fish, these enzymes seem to be involved in humoral immune response rather than in lipid metabolism, the key proteins of which were practically absent in plasma.

Reconstruction of the metabolic pathways revealed altered complement and coagulation cascades (Appendix A) upon infection progression. In addition, in the plasma of SL fish, the central enzymes of anaerobic glycolysis and pentose phosphate pathway have been determined (Appendix A). This reflects both intensifications of ATP production in severely affected fish due to substantial energy requirements of immune activation and massive release of intracellular enzymes into the plasma due to hemolysis and tissue lesion.

## 4. Discussion

### 4.1. Molecular Mechanisms of Individual Resistance to Natural V. anguillarum 

Early studies have shown that, despite the highly devastating nature of *Vibrio infection*, fish populations usually contain individuals that are resistant to this disease [6]. Thus, the original design of this experiment assumed that populations of asymptomatic trout would have specific physiological and biochemical characteristics. As expected, severely affected fish (the SL group) demonstrated the pathophysiology of systemic inflammatory response syndrome with activation of circulating granulocytes, releasing pro-inflammatory cytokines *il1b* and *il8,* and concomitant secondary organ damage (Figure 2D). In turn, according to the analysis of various physiological and immunological indicators, the metabolism in asymptomatic fish (AS group) was very similar compared to fish with clinical but not critical signs of infectious disease (IP group). Given that the individuals from the SL group with the strongest activation of the immune response demonstrated the most severe pathologies (such as furuncles and open ulcers), we suggest that they suffer from cytokine-mediated tissue damage due to over-activation of the immune system. Therefore, the fish from the IP group fight against infection while maintaining an adequate level of activation of immune reactions, which improves their survival and, presumably, makes them more resistant to the disease. What is important, *Vibrio* DNA was found in the internal organs of some individuals in both AS and IP groups, which indicates that they both were under the influence of microbial load but effectively control the infection progression.

To determine if there are more subtle metabolic differences between fish with and without tissue lesions, and to study their fine-tuning of immune regulation, the plasma of cultured trout was analyzed with proteomics. This comparison of plasma protein profiles confirmed a weak immune response and very small differences in biochemistry between asymptomatic and moderately infected fish. Compared with the IP group, the plasma of asymptomatic fish was characterized only by a significant increase in the abundance of the orthologue of the cell surface protein cadherin, which is involved in the formation of cell junctions between epithelial cells. The release of cadherin into the blood causes an increase in endothelial permeability, which promotes adhesion and migration of leukocytes into tissues [26]. The appearance of soluble cadherin in plasma may indicate destruction of the epithelium by *Vibrio* virulence factors or early degranulation of neutrophils [26,27]. Surprisingly, cadherin was absent in the plasma of trout from the IP group and was very low in fish from the SL group, suffering from endothelial lesions and hemorrhages. We can only speculate that, due to its immunomodulatory function, the high basic level of soluble cadherin in asymptomatic fish may somehow facilitate the early elimination of pathogens.

Based on previous studies, components of the complement and coagulation cascade, as well as non-cellular immunity, were suspected to be essential for trout resistance to vibriosis. In a study by Karami and colleagues [6], bathing with *V. anguillarum* caused an increase in the expression of spleen IL-4/13a, gill immunoglobulin D, and interferon-gamma in asymptomatic fish compared to the diseased ones. Hou with colleagues [4] reported an activated complement and coagulation cascade and TNF-associated immune protection in asymptomatic trout compared to untreated fish after intraperitoneal injection with *V. anguillarum*. These parameters were independent of bacterial load and have been proposed as indicators of differences between the resistant and nonresistant fish.

In our study, we did not find changes in the production of the same proteins in asymptomatic fish (AS) compared to the IP group. However, in the plasma of moderately infected fish, compared to both AS and SL groups, the abundance of the prothrombotic protein fibrinogen (both forms) was the lowest, while a prothrombotic protease inter-alpha-trypsin inhibitor heavy chain H3-like protein (ITIH3) and α-2-macroglobulin had the highest plasma levels. Noteworthy, α-2-macroglobulin production was shown to be critical to the resistance of rainbow trout to bacterial furunculosis [28]. Therefore, the observed increased levels of antiproteases in fish from the IP group can effectively suppress the dissemination of bacteria by inactivation of bacterial extracellular protease [29]. Moreover, the abundance of α-2-macroglobulin in our experiment was negatively correlated with almost all indicators of inflammation and wellbeing in trout (IPS, granulocytes, cytokines), which confirms its anti-inflammatory action and suggests its possible key role in the development of *Vibrio* pathogenesis (Appendix A.4). 

### 4.2. Molecular Characterization of Severely Affected Fish

#### 4.2.1. Serum Markers of Tissue Damage

The most notable metabolic change observed in the plasma of severely affected trout (SL) was the strong activation of the entire pathway of anaerobic glycolysis (Appendix A). A remarkable increase in the abundance of glycolytic enzymes but not the enzymes of the tricarboxylic acid cycle indicates severe metabolic acidosis and oxygen deficiency in tissues caused by massive hemorrhages [30]. Besides, activation of immune cells implies switching their metabolism to anaerobic glycolysis for rapid energy generation [31]. The appearance of cellular glycolytic enzymes in the plasma of SL trout can be an indicator of massive suicidal activation of neutrophils, as well as nonspecific hemolysis and tissue destruction in fish at the terminal stage of the disease [32].

#### 4.2.2. Antigen Recognition and Humoral Immunity Regulation

A comparison of severely affected trout (SL) and fish with the moderate and asymptomatic course of *Vibrio infection* revealed strong activation of inflammation reactions as the mechanism of *V. anguillarum* pathogenicity. In SL fish, the release of granulocytes into the bloodstream and high levels of *il1b* and *il8* were indicative of acute inflammation. Accordingly, fish from the SL group demonstrated increased plasma levels of acute phase proteins (APPs) and complement cascade actors, which regulate the innate immune response by recognition and opsonization of infectious agents [33,34].

Intelectin, haptoglobin, and hemopexin, abundant in SL fish, promote bacterial agglutination and activation of the complement system, as well as bind the iron transporters lactoferrin and heme [35]. The release of these proteins in the blood can promote iron sequestration, essential for bacterial growth [36]. Interestingly, the abundance of plasma hemoglobin was decreased in SL fish, despite *Vibrio* hemolysins being able to cause massive erythrocyte lysis [37]. This indicates efficient utilization of hemoglobin in SL fish, either by the bacteria or via the host metabolic pathways. In addition, in the case of SL trout, haptoglobin and hemopexin were not able to fulfill their known function of impending systemic inflammation in hemolytic and hemorrhagic conditions [38].

Immune proteins immunoglobulins, precerebellins, and pentraxins recognize bacterial ligands and trigger activation of complement, phagocytosis, and oxidative burst [39]. However, like classic APPs, precerebellin was increased in the plasma of SL trout, while the abundances of petraxin and immunoglobulin protein (light chain) were low in the same fish. In contrast with mammals, pentraxin in fish is known to be negatively correlated with the development of the acute phase of inflammation, indicating a specific immune regulation pathway in fish [11]. The immunoglobulin was the most abundant in trouts with a moderate course of the disease (IP group), and, thus, is presumably important for a controlled immune response during vibriosis. A similar decrease in the expression of IgDm, IgD, and IgT with an increase in the expression of AAPs was revealed in an experimental infection of trout with *V. anguillarum* [6].

A linear increase in the abundance of C-type lectin receptor B was revealed along with an increase in the severity of the disease, which indicates the importance of innate immunity in the pathogenesis of vibriosis. The receptor recognizes the bacterial antigens and triggers the activation of the complement system via the lectin pathway [40]. At the same time, binding of leukocyte cell-derived chemotaxin 2 (LECT2), increased in SL fish, to the C-type lectin receptor B can induce massive neutrophil activation in SL trout [41] leading to cytokine-mediated tissue damages.

The complement cascade in teleosts plays a critical role in innate and adaptive immunity, including pathogen recognition, opsonization, recruitment of immunocompetent cells, and elimination of pathogens [42]. Early studies showed that the complement system is involved in the resistance of rainbow trout to furunculosis [43]. In SL fish, the membrane-attacking complement complex component C7-1 was increased against the complete absence in fish from the AS and IP groups. Another isoform of this protein component C7-2 (Q6H965) was found to be expressed in all studied groups of fish (Appendix A). Importantly, in a study by Gerwick and colleagues [11], a significant increase in the expression of C7, but not other complement components, was found after dead *V. anguillarum* was administered to fish. The authors conclude that C7 synthesis may be a factor limiting the rate of activation of the entire complement cascade and is a key indicator of the health of fish infected with *Vibrio* sp. This assumption is partly supported by our results, which indicate the existence of at least some isoforms of C7 with complement regulatory function. This also confirms the assumption that the pathological course of the disease in certain individuals may be caused by hypersensitivity of innate immunity. Interestingly, a decreased expression of complement component C7 (along with increased levels of C3 and C4) was reported in asymptotic trout treated with live *V. anguillarum* compared to fish that did not contact with *Vibrio* antigens [4], while in diseased fish in this experiment the expression of C7 was significantly increased. This confirms that activation of complement trough component C7 has a negative effect on the resistance to *Vibrio* infection.

In teleosts, the nonspecific innate host defense system is more active and effective against infectious agents compared to the adaptive immune response [29]. The humoral factors of innate immunity include various degrading enzymes and antimicrobial peptides [44], such as cathelicidin 2B, ceruloplasmin, lysozyme CII, and lysozyme g, that destroy the bacterial cell wall. The levels of these proteins were significantly increased in the blood of SL fish, which is a typical response to bacterial pathogens [6]. The presence of these antimicrobial agents in the blood of fish with severe lesions may indicate the dissemination of bacteria in the internal environment of the organism. Given that cathelicidin 2B has antimicrobial activity mainly against gram-negative bacteria, its upregulation may be a protective response of humoral immunity specific to this type of infection [45].

Notably, a complex profile of differentially expressed apolipoproteins was observed in the plasma of infected trout. This protein family is extremely large (up to 17 forms can be simultaneously present in trout); many of them bind bacterial lipopolysaccharides and, apparently, have different immunomodulatory properties [35]. In our study, the abundances of apolipoproteins H, M-like (saxitoxin and tetrodotoxin-binding protein), and M were increased in the SL group (A0A060YRH9, A0A060XKK4, and A0A060YST0, respectively), whereas another isoform of apolipoprotein M (A0A060YU50) and apolipoproteins B-100 and A-I-like had the lowest plasma levels in the SL trout (Appendix A). Similar results were reported after vaccination of trout with adjuvanted hen egg-white lysozyme: the level of apolipoprotein A-IV (structurally similar to the protein A0A060YST0) increased after immunization, while the level of apolipoprotein B-100, on the contrary, decreased [35,46]. Major high-density lipoproteins apolipoproteins A-I and A-II have direct bactericidal activity [47], while the function of low-density lipoprotein component apolipoprotein B-100 in fish is not clear, but it seems to be opposite to high-density lipoproteins. Thus, the balance of various apolipoproteins in fish probably provides fine-tuning of fish immunity.

It is noteworthy that treating trout with dead or attenuated live *V. anguillarum* causes effects similar to those observed in the SL fish, namely increased expression of proinflammatory cytokines and APPs (intelectin, haptoglobin, precerebellin, LECT2, apolipoproteins, etc.) [11,46]. This suggests that the recognition of *Vibrio* antigen, rather than the microbial activity itself, can trigger a strong innate immune response, provoking a severe course of infection. It is significant that in marine species with a long evolutionary history of contact with this halophilic bacterium, the *Vibrio* antigen was reported to not cause such rapid immune response [48].

Interestingly, a positive linear correlation was found between the severity of pathology (expressed as IPS) and the abundance of antimicrobial peptide ceruloplasmin (A0A060Y0R7) and pigment epithelium-derived factor (A0A060X9D3), as well as with C type lectin receptor B (C0KIP4), apolipoprotein H (A0A060YRH9), hemopexin (A0A060XEG1) and uncharacterized protein A0A060YG41 with pentraxin domain. Therefore, the aforementioned proteins can be proposed for more detailed consideration as possible prognostic markers of the negative outcome of the disease.

#### 4.2.3. Blood Coagulation System

Another aspect of *V. anguillarum* pathogenicity is a disorder of the blood coagulation system during infection. Although the fish from the SL group had an increased content of prothrombotic fibrinogen and prothrombin (C1BHQ1), the abundance of enzymes with fibrinolytic effect angiotensin and plasminogen (A0A060WHF0) was also high compared to other groups of fish (Appendix A). At the same time, in severely damaged trout, the content of prothrombotic protease ITIH3 and α-2-macroglobulin was reduced, and blood coagulation cascade factor kininogen was not found, in contrast to plasma of trout in other groups. Similar results were reported by Hou et al., who showed activation of complement and coagulation cascades in asymptomatic trout and impairment of coagulation in diseased fish after live *V. anguillarum* injection [4]. Presumably, blood clotting could be affected by virulence factors secreted by bacteria, as it was shown that the introduction of *V. anguillarum* extracellular products provokes bleedings and necrosis [25], while the introduction of inactivated microbe, on the contrary, activates the expression of coagulation factors [46].

#### 4.2.4. Anti-Inflammatory Reactions

Compensatory anti-inflammatory reactions were also observed in severely affected fish. Increased levels of anti-inflammatory proteins, such as differentially regulated trout protein (DRTP1, CD59 glycoprotein homologue), catechol-O-methyltransferase domain-containing protein 1 (COMT), pigment epithelial-derived factor (PEDG), and actin-depolymerizing factor (gelsolin) were observed in the plasma of SL fish. Activation of the synthesis of these proteins appears to offset the harmful effects of hyperinflammation. 

### 4.3. Several Remarks for Experimental Design and Following Studies

Summarizing the results obtained, several remarks should be made regarding the design of studies on the molecular factors of trout resistance to vibriosis. Previously, Hou et al. showed that trout resistant to live *V. anguillarum* are characterized by a low level of stress accompanied by activation of immunomodulatory genes and a coagulation cascade, while diseased fish are distressed, exhibiting disturbed steroid hormone homeostasis and exacerbated immune response [4]. Our results revealed that analogous phenotype-specific responses could be revealed in both AS and IP groups of fish compared with SL fish. Since the experiments used a highly effective dose of the pathogen, the previous molecular studies [4,5] probably did not actually include fish similar to fish with a moderate course of infection, caught in a natural reservoir with a certain range of microbial load. Therefore, asymptomatic and diseased trout in previous experimental works appear to correspond to AS and SL groups rather than AS and IP groups in terms of the current research. This suggests that, in future studies, moderately infected trout should not be classified as vulnerable to infection, but, on the contrary, should be considered as potentially producing resistance factors.

At the same time, it has been previously shown that trout with a pathogenesis similar to the SL group (increased levels of cytokines and a pronounced immune response) can also survive *V. anguillarum* infection; thus, exaggerated defense response is also effective against the pathogen [6]. However, unlike the observed negative effects of immune hyperstimulation, its benefit to resistance against *V. anguillarum* is difficult to estimate due to the “survivor’s bias”, since unacceptable levels of immune system parameters cannot be measured in fish that die as a result of disease progression.

## 5. Conclusions

In this work, we describe the clinical, bacteriological, hematological, and biochemical details of pathogenesis caused by *V. anguillarum* in farmed rainbow trout. It is not yet clear to what extent the observed disorders were provoked by microbial activities or by endogenous causes induced by pathogen recognition by the fish immune system. According to the obtained results, we suggest that provoking severe inflammatory disease could be one of the virulence factors of *V. anguillarum*. Despite the powerful response of cellular immunity, complement cascade, and antimicrobial peptide production, severely damaged fish were unable to defeat the infection. Moreover, over-activation of the immune response leads to further tissue injury rather than repair. The fact that the high susceptibility to stress and enhanced inflammatory response exacerbates the course of vibriosis should be taken into account when breeding rainbow trout for disease resistance. In turn, fish presumably resistant to vibriosis demonstrated potency for early activation of the coagulation cascade, increased endothelial permeability, and enhanced production of alpha-2-macroglobulin antiprotease, which prevents bacterial dissemination into the blood. These processes can be proposed for further molecular studies of the defense mechanisms against *V. anguillarum* infection in fish.

## Figures and Tables

**Figure 2 animals-12-02642-f002:**
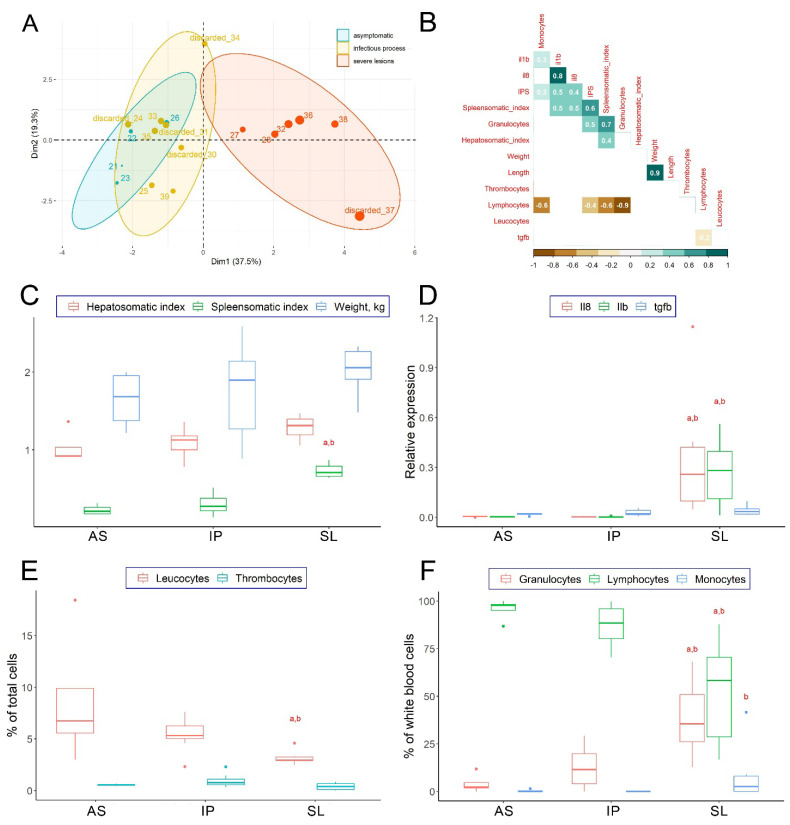
Assessment of the physiological state of rainbow trout collected on the farm during an outbreak of vibriosis (AS—asymptomatic, IP—with infectious process, SL—with severe lesions). (**A**)—principal component analysis of the measured parameters of the physiological state of trout. Samples are colored by fish group; marker size corresponds to the IPS value. Samples not included in the proteomic analysis were denoted as “discarded”. (**B**)—Spearman correlation matrix of morphophysiological parameters and biomarkers of immune response in collected rainbow trout. Only significant correlations are shown. (**C**)—morphometric parameters of trout from groups with different severity of the infectious disease. (**D**)—expression of immune-related genes in the spleen of trout from groups with different severity of the infectious disease. (**E**)—relative numbers of white blood cells in trout from groups with different severity of the infectious disease. (**F**)—types of leukocytes in the blood of trout from groups with different severity of the infectious disease. a—designates statistically significant difference from the AS group, b—designates statistically significant difference from the IP group (Wilcoxon’s two-sample test, *p* < 0.05).

**Figure 3 animals-12-02642-f003:**
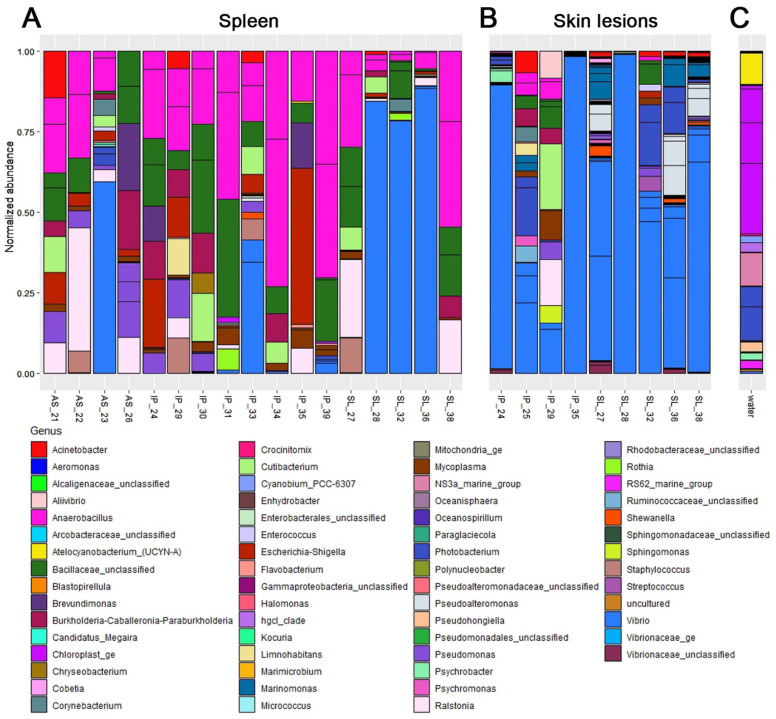
Composition of bacterial communities in rainbow trout and water collected in the fish farm during a vibriosis outbreak (AS—asymptomatic, IP—with infectious process, SL—with severe lesions). (**A**)—composition of the spleen OTUs. (**B**)—OTU composition of skin lesions (if the fish had any). (**C**)—composition of OTUs of water on the farm. OTUs are colored by genus.

**Figure 4 animals-12-02642-f004:**
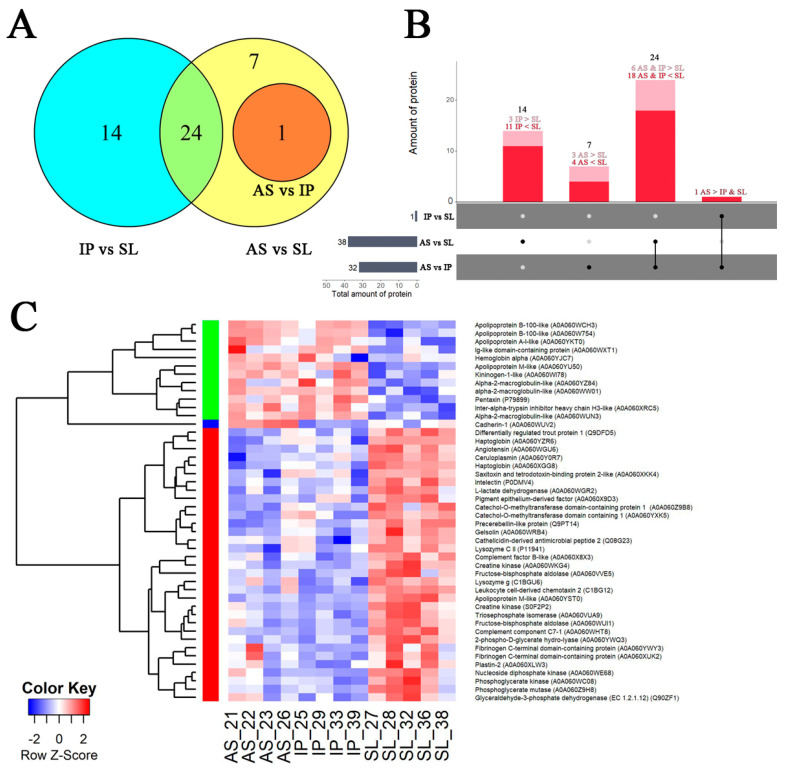
Changes in the protein composition of trout plasma in rainbow trout collected in the fish farm during an outbreak of vibriosis (AS—asymptomatic, IP—with infectious process, SL—with severe lesions). (**A**)—Venn diagram depicts overlapping and unique differentially expressed proteins between groups of fish. (**B**)—comparison of differentially expressed proteins between fish groups. (**C**)—heatmap of LFQ of trout proteins, the abundance of which was differentially expressed between groups.

## Data Availability

Photographs of collected trout supporting Appendix A are available in figshare: Borvinskaya, Ekaterina (2021): Photographs of a trout suffering from Vibrio anguillarum during an outbreak on a fish farm. Figshare. Figure. https://doi.org/10.6084/m9.figshare.17169380, accessed on 13 December 2021. All mass spectrometry data are deposited in ProteomeXchange with the identifier PXD031356. Data on the composition of bacterial communities in the spleen, skin and intestines of farmed rainbow trout available at OTU https://dor.org/10.17632/6m4jz3ywhc.2 (accessed on 13 December 2021).

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
