# Peer review of "Physiological and Biochemical Characteristics of Rainbow Trout with Severe, Moderate and Asymptomatic Course of Vibrio anguillarum Infection"

_animals, 2022, doi:10.3390/ani12192642_

Round 1
Reviewer 1 Report
The authors describe the physiological and biochemical characteristics of of Vibrio anguillarum infection in rainbow trout. Three subpopulations of fish with different severity in symptomatic damage were compared and the clinical signs of infectious processes were characterized. The manuscript is generally well written, however, I have several major points:
1. Line 113-114, line 300, “the spleen” were weighed, and the “the relative size of spleen” was used as one of the “applied biomarkers of the immune response“. As I understand, the weight and size were different measurements, what was exactly used? In addition, the absolute weight or size means little, how was the relative weight/size calculated? This should be shown in the manuscript.
2. Line 360-366, 16S rRNA amplicon seqeuncing was performed to present the microbial community in spleen and skin lesions. However,due to the low abundance of the bacteria, as well as the limitation of the 16S sequencing technology, I think it may not be accurate measurements of the microbial community,especially for the internal organ spleen as shown in Figure 3A. The conclusion in Line 365-366 needs further validation using more samples and metagenome sequencing. Moreovere, the discussion based on this result in Line 471-474 is also confusing. I suggest to delete the microbial analysis from this paper.
Reviewer 2 Report
1. It is better to state if the trout was mature or immature, and how about the gender of the experiment trout (1827 ± 471 g). Are they male, female, or mix genders? It is better to state the reason that gender is taken into consideration or why not.
2. Was the trout infected by V. anguillarum in a lab condition (artificially) or they are infected on a nature-culture farm?
3. Please state AS, IP, and SL group in the method and materials. Is there any not sacrificed trout in IP and SL groups? Did they recover or die eventually?
4. What kind of data was used for correlation analysis (Figure 2A), original data or normalized (transformed) data? If there is some process for data normalization (transformation), it is better to state. Otherwise, they do not need to state.
5. Statistical methods of figures 2C, 2E, and 2F are missing, Did they same with gene expression analysis?
6. In Proteomic analysis, several proteins showed more than one isoforms (e.g. three isoforms of alpha‐2‐macroglobulin (A0A060YZ84, A0A060WUN3 and A0A060WW01)) were identified). Is it due to alternative splicing or they came from translocation of paralogs (due to salmon/teleost specific whole genome duplication)?
7. “Activation of complement trough component C7 showed a negative effect on the resistance to Vibrio infection”. Trout exhibited duplicated genes in complement systems. Did the C7 indicate a specific C7 isoform or the total C7?
Reviewer 3 Report
The authors investigated the physiological and biochemical characteristics of rainbow trout with the severe, moderate and asymptomatic course of vibrio anguillarum infection. This manuscript (MS) was clearly written and easy to understand. This work can help the sustainability of this species farming. However, some minor issues significantly compromised the quality of this MS.
Minor comments
Abstract
· Line 15, please complete this sentence; serious problem in what?
· Line 19, please put the scientific name in parentheses.
· Line 20-21, how you realized and separated these three groups.
· Line 23, please revise this sentence.
· Line 26, Here and throughout the MS, please first mention the common name plus scientific name, and for the rest of the MS, just report the common name.
· Line 35, revise it, complement
· Line 39-40, please make sure you clarify how you define/realize the three groups.
· Here and elsewhere, report P uppercase and italic (P<0.05).
· Throughout the MS, if there is no significant difference, no need to report P-value.
· Please reorder the keywords alphabetically and capitalize each word.
· Please write the abstract more numerically about the results. You can do it by adding their numbers in parentheses.
Introduction:
• Well-developed introduction and included a clear fellow and relevant points.
· Line 49, delete the longest.
· Line 51-52, please revise it.
· Line 61-63, please revise it
· Here and throughout the MS, please first mention the common name plus scientific name, and for the rest of the MS, just report the common name.
· Please update the introduction with recent works as many studies are available from the last two years, which are not included in this section.
· Please mention the novelty of your work in the last paragraph of the introduction.
Material and methods
· Well-organized section. Clear fellow and all required details were provided.
· For each analysis, please clarify how many fish were taken.
· Line 254, Here and throughout the MS, please first mention the common name plus scientific name, and for the rest of the MS, just report the common name.
Results and discussion
· Well-written section; all necessary things have been covered.
· Line 310, please clear what biomarkers are.
· There are many interesting correlations and I suggest discussing these results.
· Figure 2B, you already labelled the groups. Please delete the name of the samples in figure 2B.
· Line 415, please revise it!!! Seriously is not a good phrase.
· Other parts were written well in this section.
· Line 495-496.
· Other parts of the discussion were well written and only required to be summarized. Please summarize the discussion section (reduce at least 10 lines).
· As a general comment: please focus on fish as hips of references and studies are available, and no need to cite other vertebrates.
When revising your manuscript, please consider all issues mentioned in the reviewers' comments carefully with clear outlines for every change made in response to their comments including suitable rebuttals for any comments you deem inappropriate. Please itemize your response to each review comment, and highlight the revised at re-submission.
Best regards
